

# Tree Growth and Water-Use Efficiency at the Himalayan Fir Treeline and lower altitudes: Roles of Climate Warming and CO₂ Fertilization

Xing Pu[1,2], Lixin Lyu[1,3*]

[1]State Key Laboratory of Vegetation and Environmental Change, Institute of Botany, Chinese Academy of Sciences, Beijing 100093, China
[2]Key Laboratory of Southwest China Wildlife Resources Conservation (Ministry of Education), China West Normal University, Nanchong 637009, Sichuan, China
[3]China National Botanical Garden, Beijing 100093, China

*Correspondence to*: Lixin Lyu (lixinlv@ibcas.ac.cn)

**Abstract.** Alpine forests are increasingly exposed to rising temperatures and intensified drought, potentially pushing species beyond their tolerance limits. However, the extent to which rising atmospheric $CO_2$ ($C_a$) can mitigate these stressors by enhancing tree intrinsic water-use efficiency (iWUE) remains unclear. We investigated the growth and physiological responses of Himalayan fir (Abies spectabilis) using basal area increment (BAI) and $\delta^{13}C$ data to track ecophysiological processes over recent decades along an elevational gradient in warming and drying sites on the Tibetan Plateau. Significant growth increases were observed at all elevations in wet regions, while negative growth trends were noted at lower elevations in dry regions. Climate–growth correlation analysis revealed that growth is primarily constrained by growing season temperatures and spring moisture availability. Tree iWUE increased over time at all elevations, with a stronger increase in wet regions. Tree growth at lower elevations in dry stands was negatively related to iWUE, whereas BAI in wet regions was positively associated with iWUE. Leaf intercellular $CO_2$ ($C_i$) increased proportionally to $C_a$ after 1965. Structural equation modeling indicated that temperature was a key driver of BAI and iWUE at all elevations in wet regions, while temperature had negative effects on BAI at lower elevations in dry regions. These results suggest that elevated $C_a$ and temperature can stimulate tree growth in high-elevation forests in wet regions, but the positive effects do not compensate for the negative impacts of reduced water availability at lower elevations in dry regions. Warming-induced drought stress may thus emerge as a more significant driver of growth compared to increasing $C_a$ levels in comparable alpine forests. Our findings provide critical insights for refining assumptions about $CO_2$ fertilization and climate change effects in ecophysiological models.

## 1 Introduction

Forests play a crucial role in regulating terrestrial carbon fluxes and influencing the rate of atmospheric $CO_2$ ($C_a$) increase, despite facing various climatic and atmospheric changes. Studies have shown that rising $C_a$ has contributed to greater tree growth, driven by the synergistic effects of warming and increased atmospheric $CO_2$ (Martínez-Sancho et al., 2018; Silva et



al., 2016; Qi et al., 2015; Saurer et al., 2014; Guo et al., 2022). However, this potential growth benefit is often overshadowed by growth declines induced by warming-related stressors, particularly drought, which has overridden the effects of rising $CO_2$ in the past decade (Peñuelas et al., 2011; Silva and Anand, 2013; Van Der Sleen et al., 2015; Liu et al., 2024; Klesse et al., 2024). Understanding the long-term physiological and growth responses of trees to global changes remains a challenge,

especially in climate-sensitive areas (Lindner et al., 2010; Charney et al., 2016; Shestakova et al., 2019; Olano et al., 2023; Sterck et al., 2024).

The Tibetan Plateau, which has experienced a rapid increase in annual mean air temperature at a rate of 0.26°C per decade over the past 40 years, is warming faster than the global average (Du, 2001). The forests of the Tibetan Plateau are particularly vulnerable to warming due to the amplified temperature increases at higher elevations (Fang and Zhang, 2019;

Guo et al., 2018; Mu et al., 2021b; Panthi et al., 2020; Sigdel et al., 2018). Although warmer temperatures have been linked to enhanced vegetation productivity (Huang et al., 2017; Piao et al., 2012; Silva et al., 2016), climate warming increases atmospheric water demand, exacerbating drought stress on plants. Water availability may therefore become increasingly critical for Tibetan Plateau forests under continued warming and rising $C_a$ (Liang et al., 2016b; Silva et al., 2016; Zhao et al., 2023). However, the long-term effects of these changes on tree physiology and growth in Tibetan Plateau forests,

particularly in relation to the unprecedented rates of modern warming and increasing $C_a$, have not been adequately addressed (Huang et al., 2017; Panthi et al., 2020; Wu et al., 2015; Xu et al., 2013).

Tree-ring records provide valuable insights into long-term physiological and growth changes (McCarroll and Loader, 2004). The isotopic discrimination against ¹³C that occurs in leaves (i.e., diffusion and carboxylation fractionations) is reflected in the stable carbon isotope ratios ($\delta^{13}C$) of the organic matter produced in a given year. There is a well-established relationship

between carbon isotopic discrimination and leaf physiology, such that $\delta^{13}C$ is directly related to assimilation rates (A) and stomatal conductance ($g_s$), which together define intrinsic water-use efficiency (iWUE) as the ratio between the two processes (Farquhar et al., 1982; Farquhar et al., 1989). Global studies using tree-ring $\delta^{13}C$ data have shown widespread increases in iWUE due to enhanced photosynthesis and carbon availability in response to rising $C_a$ (Saurer et al., 2004; Peñuelas et al., 2011; Wang et al., 2012; Keenan et al., 2013; Frank et al., 2015). However, climatic and environmental

factors such as temperature, precipitation, and nutrient availability may also influence iWUE and reduce the potential $CO_2$ fertilization effects on radial growth (Frank et al., 2015; Guerrieri et al., 2019; Liu et al., 2019; Wang et al., 2020; Zhang et al., 2018). Furthermore, the spatial variability of climate warming and decreasing moisture availability may be more influential in driving tree growth than changes in $C_a$ in cold mountain forests (Salzer et al., 2009). Recent studies have highlighted contrasting physiological strategies among plant species and elevations in response to environmental changes

(Wang et al., 2020; Garcia-Forner et al., 2016; Martínez-Vilalta and Garcia-Forner, 2017; Klein, 2014; Fang et al., 2020). Therefore, differences in water-use efficiency and ecophysiological strategies across species and regions underscore the importance of shifting from global studies to more localized, species-specific approaches for assessing the long-term effects of warming temperatures and rising $C_a$ (Voltas et al., 2020; Martínez-Sancho et al., 2018; Frank et al., 2015).





In this study, we present annually resolved iWUE and BAI data for Himalayan fir (Abies spectabilis), a widely distributed
conifer species of the Tibetan Plateau. Although some studies have explored the relationship between iWUE and growth of
Himalayan fir on the Tibetan Plateau (Huang et al., 2017; Panthi et al., 2020; Wang et al., 2020), they have typically focused
on single locations. Here, we examine a broader set of stands distributed along elevational gradients and in both wet and dry
regions of the Tibetan Plateau, aiming to identify the climatic conditions under which trees exhibit vulnerability. Our
specific objectives are: (i) to explore whether rising $C_a$ and climate changes have induced region-specific and elevation-
specific changes in tree growth and physiological parameters; (ii) to determine the extent to which changes in climate and
iWUE are related to radial growth in the study area; and (iii) to assess the physiological adjustments of trees to rising
temperature and increased atmospheric $CO_2$ over the study period.

## 2 Materials and methods

### 2.1 Site conditions and species

The study was conducted in the southern and southeastern regions of the Xizang Province, located on the southern and
southeastern Tibetan Plateau, characterized by a typical monsoon climate. The study sites spanned a broad range across the
southern and southeastern Tibetan Plateau (Fig. 1). Meteorological data, including mean temperature and precipitation, were
obtained from two meteorological stations of the China Meteorological Administration (Fig. 1) for the study period (1950s-
2010s). Instrumental records showed a mean annual temperature of 0.23°C and an average total precipitation of 429 mm in
Pali (wet region) recent decades, while the mean temperature is 2.7°C and the average total precipitation is only 282 mm in
Dingri (dry region) recent decades (Fig. S1). Additionally, we retrieved Standardized Precipitation-Evapotranspiration Index
(SPEI) data from the Climate Research Unit (CRU, University of East Anglia) TS Version 4.01 (http://climexp.knmi.nl), to
represent the water conditions of the studied forests. Himalayan fir (Abies spectabilis) is a cold-tolerant species native to the
high-elevation montane forests of the Himalayas. It has a wide elevational distribution ranging from 2,800 m a.s.l. to the
upper treeline. The species is found across the central and western Himalayas, with a significant presence on the
southwestern and southern Tibetan Plateau (Liang et al., 2016b; Panthi et al., 2020).

### 2.2 Field Sampling and Tree-Ring Width Chronologies

The tree-ring samples used in this study were collected from healthy Himalayan fir trees growing at elevations ranging from
3,378 m to 4,557 m a.s.l. in the southern and southeastern Tibetan Plateau (Table S1; Fig. 1). These sites were undisturbed
by human activities during the study period. Tree-ring increment cores were extracted from trees along an elevational
gradient in both dry (Dingjie) and wet (Shangyadong) regions. The cores were air-dried indoors, mounted on wooden slats,
and polished with progressively finer sandpaper up to 1000 grit until the tree-ring boundaries became clearly visible. Using a
microscope, tree rings from each sample were crossdated by comparing ring patterns among samples. We measured tree-ring
widths using a LINTAB 6 measuring system, with a resolution of 0.001 mm. Visual crossdating was verified with





COFECHA software (Holmes and Kozinn, 1983). All crossdated ring widths were processed using ARSTAN software to standardize the ring-width series, employing a negative exponential or linear growth curve to remove non-climatic signals. The detrended index series were then merged using the biweight robust mean method to create a standard (STD) chronology for each forest stand. For each tree-ring value, tree-ring width was converted into basal area increment (BAI, cm² per year) using the following formula proposed by Phipps and Whiton (1988), assuming balanced growth for each round (1 year = 1 round, early and late): $BAI = \pi \times (R_n^2 - R_{n-1}^2)$. Where R is the radius of the tree, and n is the year of tree-ring formation. The calculation of BAI was performed using the R package dplR (Bunn, 2008).

**2.3 Stable Carbon Isotope (δ¹³C) and Intrinsic Water-Use Efficiency (iWUE)**

Five cores from different trees were selected at each forest stand, ensuring clear and continuous ring boundaries with no missing rings. The annual rings from the five samples were pooled by year to produce a single composite isotope series for each forest stand. The wood material was ground using a centrifugal mill to ensure homogeneity and efficiency in α-cellulose extraction. The α-cellulose was extracted from the annual tree rings following standard methods (Loader et al., 1997). To maximize homogeneity, the cellulose was treated in an ultrasound unit in a hot water bath (JY92-2D, Scientz Industry, Ningbo, China) to break down the cellulose (Laumer et al., 2009). The α-cellulose was then freeze-dried for 72 hours using a vacuum freeze dryer (Labconco Corporation, Kansas City, MO, USA) prior to isotope analysis. The δ¹³C values were determined using an element analyzer (Flash EA 1112; Bremen, Germany) coupled with an isotope-ratio mass spectrometer (Delta-plus, Thermo Electron Corporation, Bremen, Germany) at the State Key Laboratory of Vegetation and Environmental Change, Institute of Botany, Chinese Academy of Sciences. The analytical errors (standard deviations) for the isotope measurements were less than 0.05‰ for δ¹³C. Calibration was performed using the International Atomic Energy Agency (IAEA) standards, USGS-24 (Graphite) and IAEA-CH3 (cellulose). All δ¹³C values are expressed relative to their respective standards (Vienna Pee Dee Belemnite for carbon isotopes and Vienna Standard Mean Ocean Water for oxygen isotopes).

The formula used to calculate δ¹³C is:

$$\delta^{13}C = \left[ \left( \frac{R_{sample}}{R_{standard}} \right) - 1 \right] \times 1000‰ ,$$ (1)

Where $R$ represents the ratio of ¹³C to ¹²C; $R_{sample}$ and $R_{standard}$ are the R values of the samples and the standard, respectively.

To accurately obtain tree-ring δ¹³C, the climate change effect, i.e., the increasing trend of atmospheric CO₂ concentration, was removed. Estimated annual atmospheric CO₂ concentration and δ¹³C values were used, derived from ice core bubble CO₂ concentrations and their δ¹³C values, along with monitoring data in recent years. The carbon isotope fractionation sequence in the tree rings was then calculated using the following equations (Farquhar et al., 1989):

$$\Delta^{13}C = \frac{\delta^{13}C_a - \delta^{13}C_p}{1 + \frac{\delta^{13}C_p}{1000}},$$ (2)





Where $^{13}C_p$ and $^{13}C_a$ were $^{13}C$ values of plant photosynthetic products and atmospheric $CO_2$, respectively. The ratio of $C_i$ to $C_a$ was calculated using:

$$\frac{C_i}{C_a} = \frac{\Delta^{13}C - a}{b - a},$$ (3)

Where $C_i$ and $C_a$ represent the concentrations of $CO_2$ in the leaves and atmosphere, respectively. a and b are constants representing $CO_2$ isotope fractionation during stomatal diffusion (4.4‰) and RuBP enzyme carboxylation (27‰). iWUE was then estimated using Ci and Ca following Ehleringer (1993):

$$iWUE = \frac{A}{g_s} = \frac{C_a - C_i}{1.6},$$ (4)

Where 1.6 is the ratio of diffusivities of water and $CO_2$ in air.

**2.4 Theoretical Gas-Exchange Scenarios**

Under rising $C_a$, tree assimilation rates (A) are expected to increase, while stomatal conductance ($g_s$) is anticipated to decrease (Franks et al., 2013). Trees regulate their stomatal aperture to either maximize carbon gain (higher A) or minimize transpiration loss (lower $g_s$). The temporal trends of $C_i/C_a$, which reflect the relationship between carbon uptake and atmospheric $CO_2$, were compared to three theoretical gas-exchange scenarios during $CO_2$ diffusion through stomata (Saurer et al., 2004). These scenarios differ in the degree to which $C_i$ follows changes in $C_a$: (i) no change ($C_i$ = constant), (ii) proportional changes ($C_i/C_a$ = constant), or (iii) equal rate changes ($C_a$ - $C_i$ = constant). Scenario 1 assumes that $C_i$ remains constant, indicating reduced $C_i/C_a$ ratios due to strong stomatal closure. Scenario 2 reflects a proportional regulation of $C_i$ by photosynthesis and stomatal conductance, maintaining a constant $C_i/C_a$ ratio. Scenario 3 assumes that $C_i$ follows the increase in $C_a$, resulting in a relatively weak stomatal response and an increasing $C_i/C_a$ ratio (Frank et al., 2015; Voelker et al., 2016).

**2.5 Statistical Analyses**

Linear trends for annual climatic variables and tree growth variables were calculated using least-squares regressions. The relationship between the $C_i$ and $C_a$ trends obtained from δ¹³C and the three theoretical gas-exchange scenarios was quantified using root mean square error (RMSE) and mean absolute error (MAE) for each study site. Mann–Kendall trend tests were conducted to identify the most recent significant warming period (1965 to the present). Structural Equation Modeling (SEM) was used to assess the effects of climate factors and iWUE on BAI over the past 40 years. SEM models were fitted using the psem function in the *pSEM* package (Lefcheck, 2016) with R version 4.1 (R Core Team, 2022). A piecewise structural equation model (pSEM) was applied to evaluate the (R²) contributions of key factors and random effects in BAI and iWUE using a linear mixed model structure (Nakagawa and Schielzeth, 2013). The overall goodness of the piecewise structural equation model was assessed using Fisher's C test, with statistical significance considered at $p > 0.05$.





## 3 Results

### 3.1 Temporal Variability in BAI, iWUE, and Cellulose Stable Carbon Isotopes

The basal area increment (BAI) of Himalayan fir exhibited a significant increasing trend at the treeline in the dry region and at all elevations in the wet region after 1965 (Fig. 2). In contrast, no significant trend was detected at the middle elevation, and a decreasing growth trend was observed at the lower elevations in the dry region (Fig. 2b). Similarly, the intrinsic water-use efficiency (iWUE) of Himalayan fir increased significantly, with a steep rise after 1965 (Fig. 3b). The carbon isotopic composition ($\delta^{13}C$) of tree rings showed a clear decreasing trend over recent decades, reflecting the rising atmospheric $CO_2$

concentration ($C_a$) across all elevations in both dry and wet regions (Fig. S2). Our results indicate that tree growth increased with the rising iWUE in wet regions, while the relationship between iWUE and growth shifted from a significant positive correlation to a negative correlation as elevation decreased in the dry regions (Fig. 4).

### 3.2 Climate Responses of BAI

The BAI of Himalayan fir exhibited region-specific climate sensitivity, with consistent responses observed in the wet region
and varying responses along the elevation gradient in the dry region. In the wet region, tree growth showed a significant positive correlation with both the previous autumn temperature and the growing season (June to September) temperature at all elevations (Fig. 5). In the dry region, however, tree growth was positively correlated with the previous autumn and growing season temperature at the treeline, while a significant negative correlation was observed at lower elevations (Fig. 5). Additionally, tree growth in the wet region was positively correlated with spring precipitation, whereas in the dry region,
spring precipitation was negatively correlated with tree growth at lower elevations (Fig. 5).

### 3.3 Theoretical Scenarios of $C_i/C_a$

A comparison of the $\delta^{13}C$-based $C_i/C_a$ records with three theoretical gas-exchange scenarios revealed distinct patterns across different elevations (Fig. 6). In the wet region, time series of the $C_i/C_a$ ratio showed marginally significant increasing trends over time at the treeline and middle elevation, but a decreasing trend at lower elevations (Fig. 6a). In the dry region, the $C_i/C_a$
ratio increased over time at the treeline and lower elevations, with a decreasing trend observed at middle elevation. Statistically, the relationship was marginally significant at the treeline and lower elevations (Fig. 6b). The $C_i/C_a$ ratios largely followed the "$C_i$ = constant" scenario (Scenario 1) in the wet region, indicating a strong stomatal response. At the treeline in the dry region, the $C_i/C_a$ ratio remained constant throughout the study period, while for the middle and lower elevations in the dry region, the $C_i/C_a$ ratios also remained stable (Fig. 6; Table S2).

### 3.4 Factors Regulating Changes in BAI

Multiple regression models identified the contributions of temperature, precipitation/moisture availability, and iWUE to tree growth over the past decades (Fig. 7). In the wet region, the previous autumn temperature (Tgrs) and the growing season



temperature (Tpaut) explained most of the variation in tree growth, with additional contributions from iWUE and spring precipitation at lower elevations (Fig. 7). In the dry region, tree growth was primarily influenced by spring moisture availability and temperature (Fig. 7). We concluded that these variables explained 25-58% of the variation in tree growth, as revealed by piecewise structural equation modeling (pSEM) (Fig. 8). The pSEM analysis indicated that Tgrs was the critical factor affecting BAI in the wet region (Fig. 8a). Additionally, a significant positive direct effect of iWUE on BAI was observed at lower elevations in the wet region (Fig. 8). Tgrs was also shown to drive iWUE patterns in the wet region, with a significant positive influence (Fig. 8). In the dry region, Tgrs and spring temperature (Tspr) had a significant negative effect on BAI at middle and lower elevations. However, BAI was positively correlated with spring moisture availability (RHspr and Pspr) at the treeline and middle elevations (Fig. 8).

## 4 Discussion

### 4.1 Tree Growth and Its Climatic Responses

Our study reveals a distinct pattern of tree growth across different regions and elevations on the Tibetan Plateau, with a strong acceleration of growth in the wet region and at higher elevations in the dry region, contrasting with decreasing growth trends at lower elevations in the dry region. These results diverge from the widely observed global trend of declining tree growth and increased forest mortality due to high temperatures and drought, particularly in other parts of the world (Hartmann et al., 2018; Mirabel et al., 2023; Allen et al., 2010). While numerous studies have documented the detrimental effects of warming and drought on forest ecosystems, particularly in the northeastern Tibetan Plateau (Liang et al., 2016a; Wang et al., 2018), our findings suggest that the Tibetan Plateau, particularly in its southeastern and southern regions, is undergoing a phase of warming and humidification that has benefited tree growth (Shi et al., 2020; Mu et al., 2021a; Guo et al., 2022).

The acceleration of growth in the wet regions aligns with the well-established finding that increased temperature can enhance tree productivity in humid climates (Liang et al., 2016b; Wang et al., 2023; Silva et al., 2016), which is further confirmed by our results showing a positive correlation between growth and growing season temperature. However, this contrasts with the negative or stable growth trends observed in the dry region, particularly at lower elevations, where warming-induced drought stress appears to inhibit growth. Our study extends existing research by highlighting that tree growth in moisture-limited ecosystems (such as the dry regions) is more constrained by water availability than by warming. This reinforces the growing body of literature suggesting that $CO_2$ fertilization effects are more likely to occur in temperature-limited ecosystems (Körner, 2015), whereas moisture-limited regions are less responsive to increased $C_a$ (Wu et al., 2015).

This research provides novel insights into how specific regional climatic shifts, particularly warming and humidification, may lead to divergent growth responses in montane forests. In contrast to the increasing mortality rates and growth decline observed in other high-elevation forest systems globally (Lévesque et al., 2014; Linares & Camarero, 2012), our study





suggests that areas undergoing warming and increased moisture availability could experience enhanced tree growth, provided drought stress remains manageable.

**4.2 Specific Variability of Isotope-Based Ecophysiological Parameters**

Our results show that Himalayan fir exhibited a consistent decrease in $\delta^{13}C$ values, leading to a significant increase in iWUE, particularly in the wet region and at the treeline in the dry region. These findings provide a clear physiological signature of

how trees are adjusting their water use efficiency in response to both rising temperatures and increasing atmospheric $CO_2$. In the wet region, Himalayan fir maintained relatively constant intercellular $CO_2$ ($C_i$) levels, while discrimination against $^{13}C$ was higher, resulting in increased iWUE. This suggests a strategic drought-tolerance mechanism, enabling the species to balance transpiration and carbon assimilation effectively (Aranda et al., 2000; Panthi et al., 2020).

Our findings are consistent with regional and global studies that have reported increased iWUE in response to rising $C_a$

(Frank et al., 2015; Guerrieri et al., 2019; Huang et al., 2017). However, the novelty of our study lies in the detailed analysis of how different regions on the Tibetan Plateau exhibit variable iWUE responses, especially under contrasting climatic conditions. The maintenance of constant $C_i/C_a$ ratios in the dry region at the treeline suggests a dynamic gas exchange strategy, with trees adjusting their stomatal conductance in response to changing environmental conditions (Saurer et al., 2004; Walker et al., 2015). This provides new insights into how trees in moisture-limited ecosystems may adopt different

physiological strategies compared to those in more humid regions. In this regard, our study highlights the spatial variability in leaf-level responses to rising $C_a$, offering a deeper understanding of species-specific ecophysiological adaptations to climate change.

**4.3 Physiological Responses to Rising $CO_2$**

The response of Himalayan fir to rising atmospheric $CO_2$ ($C_a$) in our study is more complex than previously reported, with

species-specific and site-specific variations in physiological responses. While many studies have suggested that increasing $C_a$ leads to higher photosynthetic rates and greater water-use efficiency, this is not a uniform response (Saurer et al., 2014; Shestakova et al., 2019). Our study builds on this foundation by showing that the effect of $C_a$ is modulated by other factors, such as temperature, drought, and nutrient availability, which vary spatially across the Tibetan Plateau. Our research underscores the need for more localized studies to better understand the spatial and temporal variations in tree responses to

rising $C_a$, particularly in regions where temperature and moisture availability interact in complex ways.

Our findings that Himalayan fir exhibits near-constant $C_i/C_a$ ratios in the wet region and at higher elevations in the dry region suggest a proportional regulation of carbon assimilation and stomatal conductance, aligning with the "$C_i$ = constant" scenario observed in many conifers (Frank et al., 2015; Saurer et al., 2004). This supports the hypothesis that trees can dynamically adjust their gas exchange to optimize both carbon gain and water conservation (Voelker et al., 2016; Walker et al., 2015).

The dynamic nature of leaf gas exchange strategies in response to rising $C_a$ represents a novel contribution of our study,

emphasizing that trees are not passive responders but actively modulate their physiological traits to cope with changing environmental conditions.

## 4.4 Tree Growth Relationships to iWUE and Climate

Our study also highlights the critical relationship between tree growth and iWUE, particularly in the context of regional climatic variations. The positive correlation between iWUE and basal area increment (BAI) in the wet region and at higher elevations, and the more limited response in the dry region, underscores the role of climate in shaping tree growth patterns. Our findings align with global studies that report variable growth trends in response to rising $CO_2$, with some regions showing positive growth responses while others exhibit little to no effect (Peñuelas et al., 2011; Silva and Anand, 2013). However, the novelty of our work lies in the recognition that growth responses are not only driven by $CO_2$ fertilization but are also influenced by complex interactions with temperature and moisture availability. The negative correlation between BAI and iWUE in the dry region, particularly at lower elevations, supports the idea that drought stress can offset the benefits of $CO_2$ fertilization, which has been observed in other studies (Fang et al., 2020; Granda et al., 2014).

Our study provides new insights into how the combined effects of rising $CO_2$ and temperature, as well as changing moisture availability, influence tree growth and carbon cycling on the Tibetan Plateau. The significant increase in iWUE, combined with positive temperature effects, contributes to the observed growth trends in wet regions, highlighting the importance of local climatic conditions in determining tree productivity. This adds to the growing body of research suggesting that spatial variability in climate change impacts is a key factor driving tree growth responses (Salzer et al., 2009; Körner, 2003).

## 5 Conclusions

In summary, our study provides novel insights into the ecophysiological and growth responses of Himalayan fir to climate change on the Tibetan Plateau. By examining the effects of rising atmospheric $CO_2$, temperature, and drought across different regions and elevations, we have shown that climate change has diverse effects on tree growth, with moisture availability being a critical limiting factor at lower elevations. Our findings underscore the importance of considering spatial and regional differences when assessing the impacts of climate change on forest ecosystems, and highlight the complex interactions between temperature, moisture, and $CO_2$ in shaping tree growth patterns. These insights are crucial for refining our understanding of forest dynamics and carbon cycling in montane ecosystems, and for predicting the future trajectory of high-elevation forests under climate change.



**Author contribution**

L.L. acquired the funding and designed the study. X.P. processed the tree-ring samples, conducted the analysis and wrote the first draft. L.L. and X.P. interpreted the results, revised the manuscript, and contributed to writing. Both authors reviewed
and approved the final submission.

**Acknowledgments**

This research was funded by the Natural Science Foundation of China (42271074) and the Doctoral Scientific Research Foundation of China West Normal University (412999). We are grateful to Qi-Bin Zhang for his supervision on the research. We also want to thank Hongyan Qiu for her kind assistance on the crossdating of the tree-ring samples.

**Declaration of competing interest**

The authors declare no competing interests.

**Data availability**

Data will be made available on request.

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

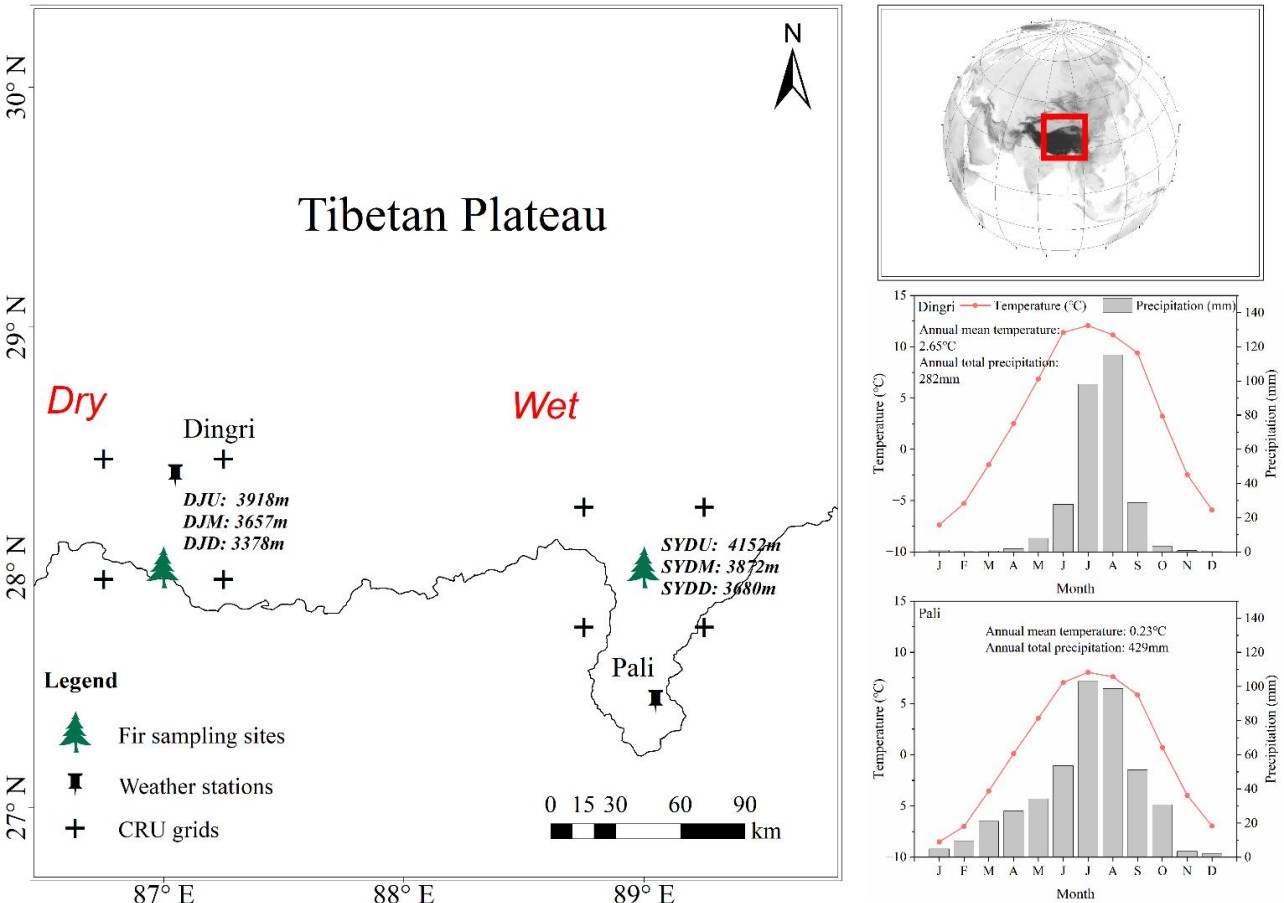

**Figure 1. Location of the study sites and weather stations for Himalayan fir on the Tibetan Plateau. Climate diagrams are based on meteorological records from the Dingri and Pali weather stations for the period 1965–2013.**



**Figure 2. Temporal trends of basal area increment (BAI) for Himalayan fir at different altitudes during the period 1900−2010s. The symbol "β" represents the slope of BAI (cm² year⁻¹).**





**Figure 3. Temporal trends of intrinsic water-use efficiency (iWUE) during 1900-2010s for Himalayan fir at different altitudes. The symbol "β" represents the slope of iWUE (μmol mol⁻¹).**





**Figure 4. Relationships between intrinsic water-use efficiency (iWUE) and basal area increment (BAI) of Himalayan fir at different altitudes across two hydrologically distinct sites.**





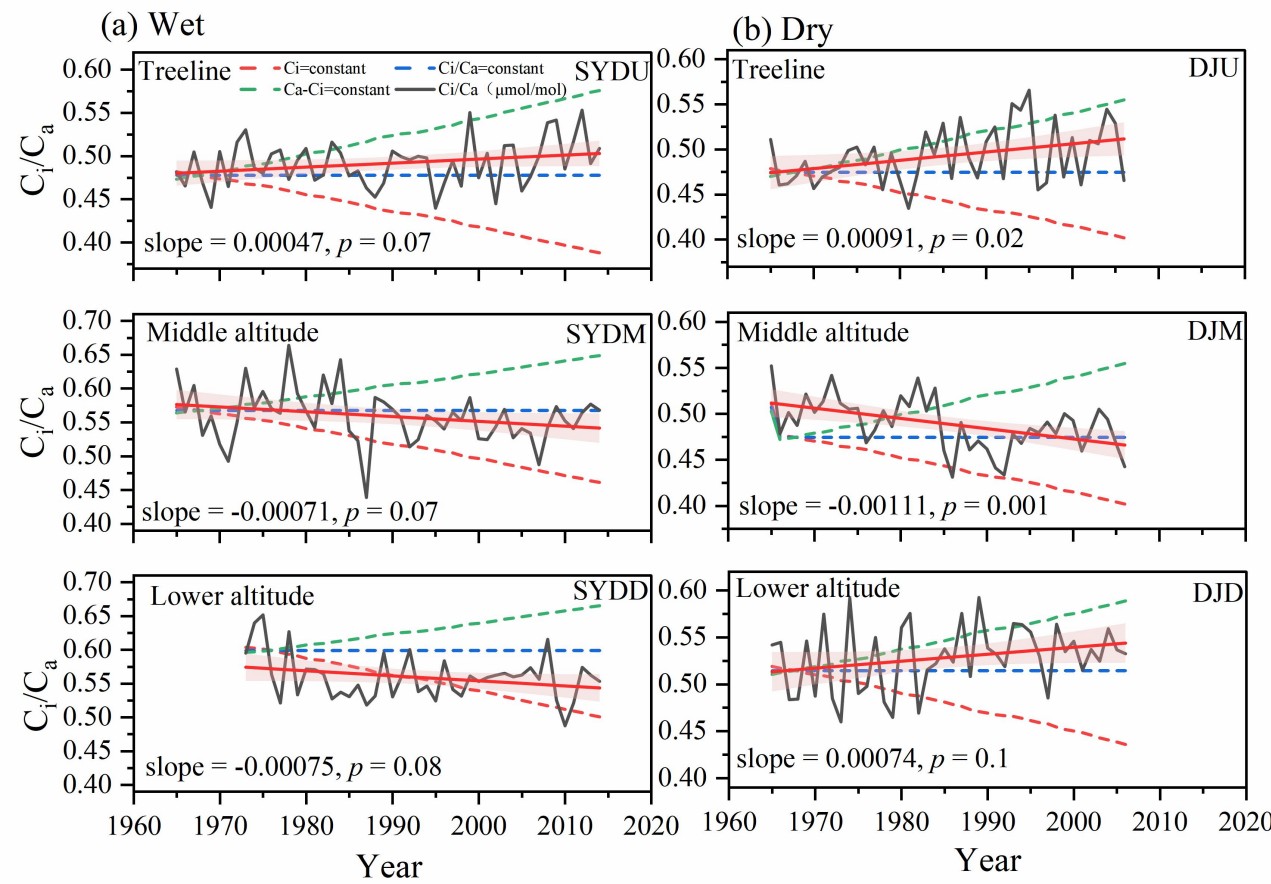

**Figure 5. Long-term changes in the ratio of intercellular to atmospheric $CO_2$ ($C_i/C_a$) for Himalayan fir during the period 1965–2013 at different altitudes. $C_i/C_a$ values under different scenarios were calculated based on theoretical models of plant gas exchange regulation in response to rising atmospheric $CO_2$: (1) $C_i$ = constant, (2) $C_i/C_a$ = constant, and (3) $C_a$ - $C_i$ = constant.**





**Figure 6. Pearson correlation coefficients between basal area increment (BAI) and climatic variables, including (a) mean temperature and (b) total precipitation, on both monthly and seasonal scales for Himalayan fir during the period 1965–2013. "Grs" denotes the growing season (June to September). "Cor_value" and "abs_value" represent the correlation coefficient value and the absolute value of the correlation, respectively.**





**Figure 7. Effects of multiple environmental factors on fir growth at different altitudes. The symbols #, *, **, and *** denote significance levels at p<0.1p<0.1, p<0.05p<0.05, p<0.01p<0.01, and p<0.001p<0.001, respectively. Abbreviations: Tspr = spring mean temperature; Tgrs = growing season mean temperature (June to September); RHspr = spring relative humidity; RHgrs = growing season relative humidity (June to September); RHwin = winter relative humidity; SPEIspr = spring standardized precipitation-evapotranspiration index; SPEIgrs = growing season standardized precipitation-evapotranspiration index (June to September).**





**Figure 8. Piecewise structural equation meta-model (pSEM) used to assess the influences of climatic factors on basal area increment (BAI) and intrinsic water-use efficiency (iWUE) of Himalayan fir during the period 1965–2010s at different altitudes. Numbers adjacent to each arrow indicate the standardized regression coefficients for each path. Arrow thickness represents the strength of the effect, while color indicates the direction (red = positive, blue = negative). Solid arrows denote significant effects, whereas dashed arrows denote nonsignificant effects. Symbols #, *, **, and *** indicate significance levels at p < 0.1, p < 0.05, p < 0.01, and p < 0.001, respectively. Abbreviations: Tpaut = previous autumn mean temperature; Tsum = summer mean temperature; Tspr = spring mean temperature; Tgrs = growing season mean temperature (June to September); RHspr = spring relative humidity; RHgrs = growing season relative humidity (June to September); RHwin = winter relative humidity; SPEIspr = spring standardized precipitation-evapotranspiration index; SPEIgrs = growing season standardized precipitation-evapotranspiration index (June to September).**