# Peer review of "Tree Growth and Water-Use Efficiency at the Himalayan Fir Treeline and lower altitudes: Roles of Climate Warming and CO2 Fertilization"

_EGUsphere, 2025_

## Author Comment (AC1)

*RC1: 'Comment on egusphere-2025-952', Anonymous Referee #1, 14 Apr 2025*

*This study investigates the physiological and growth responses of Himalayan fir to climate warming and $CO_2$ fertilization across elevational gradients in wet and dry regions of the Tibetan Plateau. The research addresses a critical knowledge gap regarding the interplay between $CO_2$ fertilization, temperature, and drought stress in high-elevation forests. While the manuscript presents valuable insights, several issues need clarification to strengthen the conclusions.*

**[Response]:** We are very grateful to the reviewer for the insightful and constructive comment on our paper. We have addressed the comments and concerns by carefully responding to each of the comments.

*Major Concerns:*

*1. Interpretation Issue of Ci/Ca Scenarios: The comparison of observed Ci/Ca trends to theoretical scenarios lacks quantitative validation. The statement that Ci/Ca "largely followed Scenario 1" is qualitative and not statistically tested. Please quantify deviations from theoretical scenarios using goodness-of-fit metrics (e.g., RMSE, AIC) and report significance tests.*

**[Response]:** We appreciate the reviewer's insightful comment regarding the statistical tests for the theoretical treatment of Ci/Ca in our initial analysis. We acknowledge that these models were oversimplified and failed to provide meaningful mechanistic explanations for our core findings. As this limitation was also noted by Reviewer #2, we have removed both the Ci/Ca analysis and related discussion from the revised manuscript.

*2.Temporal coverage issue: The results section mentions post-1965 trends for most of the analyses, but we lack a temporal changes of temperatures to justify its relevance to modern climate change. Consider add a figure to show the climate warming in the study region.*

**[Response]:** Comment accepted. We have provided the main climate variables over the past decades to better contextualize the research. According to the climate data,

the study area has experienced a significant and continuous warming trend over the past century (Fig. S5a). The change rate of the average temperature    increased from 0.007°C/year (1901-1964) to 0.032°C/year (1965-2013) in the drier site (DJ), while the change rate of temperature in wetter site (SYD) increased from 0.008°C/year (1901-1964) to 0.023°C/year (1965-2013). However, the annual precipitation showed no significant changes since 1901 (Fig. S5b), resulting a general decreasing trend of SPEI for both sites (Fig. S5c)

[Figure]

**Fig. S5** Temporal variations of mean temperature (a), total precipitation (b), SPEI (c) for the

sampling sites based on the climate data extracted from the CRU TS 4.04 for the period 1901–2013.

***Specific Concerns:***

*L14: The species name should be in italic. The same for L64.*

**[Response]:** Thanks for your comment. Done as suggested.

*L86: Liang et al., (2016) did not analysis this tree species, please remove it from here.*

**[Response]:** Comment accepted. We have removed the reference in the revised manuscript.

*L120: The text briefly mentions that the data on atmospheric carbon dioxide concentration is derived from ice core data, but does not elaborate on the specific source. Please provide the source and the reference literature.*

**[Response]:** Thanks for your comment. Detailed information on the $CO_2$ were added in the revised manuscript: "$CO_2$ concentration data was derived from a combination of the reconstructed values (period 1900-2003) using ice cores (Mccarroll and Loader, 2004; Boucher et al., 2014) and the direct observations of $CO_2$ concentration for the period 2004-2013, which were obtained from the Mauna Loa Observatory of America (http://www.esrl.noaa.gov/gmd/obop/mlo/)."

*L217: The sections 4.2 and 4.3 should be merged and summarised by a more physiologically meaningful title.*

**[Response]:** We sincerely appreciate your insightful comments, which have helped strengthen our manuscript. In response to your suggestions, we have consolidated Sections 4.2 and 4.3 into a more focused section titled: "4.2 Spatial heterogeneity of tree growth-iWUE relationships ".

Specifically, we firstly removed the discussions on the oversimplified theoretical models of Ci/Ca changes, as they did not provide meaningful mechanistic explanations, then sharpened our discussion to emphasize the potential physiological

mechanisms driving the spatial heterogeneity in BAI-iWUE relationships as modulated by the interaction between altitudinal and moisture gradients. These revisions have allowed us to present a more coherent and mechanistically grounded interpretation of our main findings.

**Reference**

Boucher, E., Guiot, J., Daux, V., Danis, P. A., and Dussouillez, P.: An inverse modeling approach for tree-ring-based climate reconstructions under changing atmospheric $CO_2$ concentrations, Biogeosciences,11,12(2014-06-17), 10, 3245-3258, 2014.

McCarroll, D. and Loader, N. J.: Stable isotopes in tree rings, Quaternary Science Reviews, 23, 771-801, 10.1016/j.quascirev.2003.06.017, 2004.

Again, thank you very much for your insightful comments. We believe these revisions have substantially elevated the quality of our work. Should any additional clarifications be needed, we would be happy to address them.

---

## Author Response (AR1)

*Co-editor-in-chief decision*:

*The authors have revised the manuscript thoroughly, and it now meets the standards of Biogeosciences. Please revise the manuscript once more in accordance with the comments from the Associate Editor.*

**[Response]:** We sincerely appreciate the co-editor-in-chief Frank Hagedorn's careful consideration of our manuscript. We are grateful to the associate editor and the two anonymous reviewers for their insightful comments and constructive suggestions, which have significantly improved the quality of our paper. In response to all comments, we have thoroughly revised the manuscript accordingly. Below, we provide a point-by-point response to each comment, detailing the changes we have made.

*Associate editor comments:*

*After carefully considering the concerns and comments raised by the two anonymous reviewers, I have evaluated your revised manuscript positively. Overall, I am satisfied with the improvements made in this version. I recommend that you perform a thorough check for any remaining typographical errors and carefully edit the language for clarity and consistency.*

**[Response]:** We sincerely appreciate the associate editor Matteo Garbarino's positive evaluation of our revised manuscript and the constructive feedback. In accordance with your recommendation, we have conducted a thorough proofreading of the manuscript to eliminate any typographical errors. Additionally, we have carefully refined the language to improve clarity and consistency throughout the paper. We believe these refinements further strengthen the manuscript and appreciate the opportunity to submit this improved version for your consideration.

**RC1: 'Comment on egusphere-2025-952', Anonymous Referee #1, 14 Apr 2025**

*This study investigates the physiological and growth responses of Himalayan fir to climate warming and $CO_2$ fertilization across elevational gradients in wet and dry regions of the Tibetan Plateau. The research addresses a critical knowledge gap*

*regarding the interplay between $CO_2$ fertilization, temperature, and drought stress in high-elevation forests. While the manuscript presents valuable insights, several issues need clarification to strengthen the conclusions.*

**[Response]:** We are very grateful to the reviewer for the insightful and constructive comment on our paper. We have addressed the comments and concerns by carefully responding to each of the comments.

*Major Concerns:*

*1. Interpretation Issue of Ci/Ca Scenarios: The comparison of observed Ci/Ca trends to theoretical scenarios lacks quantitative validation. The statement that Ci/Ca "largely followed Scenario 1" is qualitative and not statistically tested. Please quantify deviations from theoretical scenarios using goodness-of-fit metrics (e.g., RMSE, AIC) and report significance tests.*

**[Response]:** We appreciate the reviewer's insightful comment regarding the statistical tests for the theoretical treatment of Ci/Ca in our initial analysis. We acknowledge that these models were oversimplified and failed to provide meaningful mechanistic explanations for our core findings. As this limitation was also noted by Reviewer #2, we have removed both the Ci/Ca analysis and related discussion from the revised manuscript.

*2.Temporal coverage issue: The results section mentions post-1965 trends for most of the analyses, but we lack a temporal changes of temperatures to justify its relevance to modern climate change. Consider add a figure to show the climate warming in the study region.*

**[Response]:** Comment accepted. We have provided the main climate variables over the past decades to better contextualize the research. According to the climate data, the study area has experienced a significant and continuous warming trend over the past century (Fig. S2a). The change rate of the average temperature increased from 0.007 ℃/yr (1901-1964) to 0.032 ℃/yr (1965-2013) in the drier site (DJ), while the change rate of temperature in wetter site (SYD) increased from 0.008 ℃/yr

(1901-1964) to 0.023 °C/yr (1965-2013). However, the annual precipitation showed no significant changes since 1901 (Fig. S2b), resulting a general decreasing trend of SPEI for both sites (Fig. S2c).

We have added the related descriptions in the main text as follows: "The CRU data reveal a significant warming trend in the study area over the past century, with the rate of temperature increase accelerating from 0.007 °C/yr (1901–1964) to 0.032 °C/yr (1965–2013) at the drier site (DJ) and from 0.008 °C/yr to 0.023 °C/yr at the wetter site (SYD) (Fig. S2a). In contrast, annual precipitation remained stable since 1901 (Fig. S2b), leading to an overall decline in SPEI at both sites (Fig. S2c).".

[Figure]

**Figure. S2 Temporal variations of mean temperature (a), total precipitation (b), SPEI (c) for the sampling sites based on the climate data extracted from the CRU TS 4.04 for the period 1901–2013.** The dashed lines depict piecewise linear regression models, with a breakpoint at 1965. The model statistics include coefficient of determination ($R^2$) and associated p-values.

The dashed lines represent piecewise linear regressions, with a breakpoint at 1965.

***Specific Concerns:***

*L14: The species name should be in italic. The same for L64.*

**[Response]:** Thanks for your comment. Done as suggested.

*L86: Liang et al., (2016) did not analysis this tree species, please remove it from here.*

**[Response]:** Comment accepted. We have removed the reference in the revised manuscript.

*L120: The text briefly mentions that the data on atmospheric carbon dioxide concentration is derived from ice core data, but does not elaborate on the specific source. Please provide the source and the reference literature.*

**[Response]:** Thanks for your comment. Detailed information on the $CO_2$ were added in the revised manuscript: "The atmospheric $CO_2$ concentration data was derived from a combination of the reconstructed values (period 1900-2003) using ice cores (Mccarroll and Loader, 2004; Boucher et al., 2014) and the direct observations of $CO_2$ concentration for the period 2004-2013, which were obtained from the Mauna Loa Observatory of America (http://www.esrl.noaa.gov/gmd/obop/mlo/)."

*L217: The sections 4.2 and 4.3 should be merged and summarised by a more physiologically meaningful title.*

**[Response]:** We sincerely appreciate your insightful comments, which have helped strengthen our manuscript. In response to your suggestions, we have streamlined and consolidated Sections into a more focused section titled: "4.2 Spatial heterogeneity of tree growth-iWUE relationships ".

Specifically, we firstly removed the discussions on the oversimplified theoretical models of Ci/Ca changes, as they did not provide meaningful mechanistic explanations, then sharpened our discussion to emphasize the potential physiological mechanisms driving the spatial heterogeneity in BAI-iWUE relationships as

modulated by the interaction between altitudinal and moisture gradients. These revisions have allowed us to present a more coherent and mechanistically grounded interpretation of our main findings.

The revised section is as follows:

"Our findings reveal a consistent decline in $\delta^{13}C$ values in Himalayan fir, corresponding to a marked increase in intrinsic water-use efficiency (iWUE), particularly in mesic regions and at treeline sites in arid zones. This trend aligns with global observations of enhanced iWUE under rising atmospheric $CO_2$ ($C_a$) (Frank et al., 2015; Guerrieri et al., 2019; Huang et al., 2017). However, our results demonstrate greater complexity in the iWUE response of Himalayan fir than previously documented, with pronounced site- and altitude-specific variability. Specifically, at the mid- and low-altitude sites in arid region, limited moisture availability appears to constrain iWUE gains despite rising $C_a$, whereas the forest stands with more moisture availability, including wet sites and the treeline site of the drier region, exhibit more pronounced iWUE increases. These findings provide a physiological framework for understanding how trees adjust water-use strategies under concurrent increases in $CO_2$ and temperature.

While elevated $C_a$ is frequently associated with enhanced photosynthesis and iWUE (Saurer et al., 2014), the observed increase in intrinsic water-use efficiency (iWUE) does not necessarily correspond to enhanced radial growth, as evidenced by the non-significant relationship between basal area increment (BAI) and iWUE, particularly in the dry regions (Peñuelas et al., 2011; Silva and Anand, 2013). This pattern suggests a drought-tolerance strategy that optimizes the trade-off between transpiration and carbon assimilation, highlighting the importance of local-scale ecophysiological adaptations to climate change (Aranda et al., 2000; Panthi et al., 2020). The negative correlation between BAI and iWUE at the low-altitude stands in the dry region supports the idea that drought stress can offset the benefits of $CO_2$ fertilization (Fang et al., 2020; Granda et al., 2014). Moreover, previous investigations have revealed altitudinal divergence in the physiological mechanisms driving iWUE enhancement: while treeline populations primarily achieved increased

iWUE through photosynthetic enhancement, populations at lower elevations predominantly relied on stomatal conductance reduction (Pu & Lyu, 2023). In this study, divergent intercellular $CO_2$ concentrations ($C_i$) across sites suggest plasticity in leaf gas exchange (Fig. S5), demonstrating that trees actively regulate physiological traits in response to rising $C_a$, rather than responding passively (Voelker et al., 2016; Walker et al., 2015). These adjustments reflect an adaptive capacity to balance carbon gain with water conservation under changing climatic conditions.

These findings emphasize the need for meso- to local-scale investigations to unravel the interactive effects of $C_a$, temperature, and moisture on tree physiology. This is particularly critical in high-altitude ecosystems like the Himalayas, where climatic gradients create complex, non-linear responses in tree growth and iWUE (Salzer et al., 2009; Körner, 2003). Future research should prioritize mechanistic models that integrate these spatial and temporal variations to improve predictions of forest dynamics under climate change."

*General comments*

*This manuscript examines the long-term physiological and growth responses of Abies spectabilis across elevational gradients in wet and dry regions of the Tibetan Plateau, using tree-ring width, basal area increment (BAI), and $^{13}C$-derived intrinsic water-use efficiency (iWUE). The authors aim to disentangle the relative roles of climate warming and atmospheric $CO_2$ in modulating tree growth. The study addresses timely questions in forest ecophysiology and uses standard dendrochronological and isotope techniques. However, the manuscript suffers from several significant conceptual, methodological, and shortcomings. The analysis relies heavily on correlative patterns/scenarios with limited mechanistic interpretation. Basic sample metadata is omitted and critical information on detrending and sample replication is lacking. Moreover, while the Tibetan Plateau is an understudied region, the manuscript largely reproduces well-established findings regarding iWUE trends, growth–climate relationships, and physiological strategies under $CO_2$ enrichment. The analysis does not introduce new mechanisms, theory, or methods. Thus, its novelty lies primarily in applying standard approaches to a geographically distinct context-which is of regional interest, but not a substantial conceptual advancement for the field.*

**[Response]:** We sincerely appreciate the time and effort taken to evaluate our manuscript, and we are grateful for the constructive feedback that will undoubtedly strengthen our work. Below we address each of the main concerns raised:

**Conceptual Advancements:** Our study advances current understanding by revealing how the interplay between elevational and moisture gradients generates divergent physiological responses in high-elevation forests dominated by a single species (*Abies spectabilis*). Importantly, we identified critical moisture-dependent thresholds in $CO_2$ response, demonstrating that even within the same species, trees exhibit fundamentally different physiological strategies depending on their local environmental context. These findings provide compelling empirical evidence from a climate-sensitive yet understudied region that challenges prevailing assumptions

about uniform species-level responses to climate change in ecosystem models.

**Methodological Concerns:** We have substantially revised the Methods section to provide comprehensive information on the samples, including detailed descriptions of detrending approaches and sample statistics (Table S1-S2). In terms of mechanistic interpretation, we have provided a table to detail the parameters and stats of the SEMs (Table S3) .

Below, please see a detailed point-by-point response to your concerns.

*Major concerns*

*1. The authors fail to report key information about the sampled trees, such as sample sizes per site, tree ages (range and mean), diameter at breast height (DBH), or inter-individual variability. Given the long temporal scope, this omission undermines the interpretation of growth trends. Tree age can confound long-term growth patterns and sensitivity to climate or CO₂. This basic metadata must be included in a table or appendix.*

**[Response]:** Thanks for your comment. We fully agree that these basic metadata are important for the interpretation on the results. Therefore, we have added more information on the tree ages and DBH in the Table S1. Moreover, we have also provided the specific information on the tree cores that were used for isotope analysis in the Table S2.

**Table S1. Information for the tree-ring samples on the southern Tibetan Plateau.**

| Position | Site | Species | Latitude (°N) | Longitude (°E) | Altitude (m) | Trees | DBH (cm) | Mean age | No. of stems/ha | TRW time span | rbar | MS |
|---|---|---|---|---|---|---|---|---|---|---|---|---|
| Treeline | SYDU | *ABSP* | 27.506 | 88.99 | 4152 | 70 | 31.0±21.5 | 68 | 146 | 1784-2014 | 0.41 | 0.18 |
| Middle altitude | SYDM | *ABSP* | 27.515 | 88.99 | 3872 | 31 | 28.5±10.0 | 47 | 840 | 1837-2014 | 0.34 | 0.19 |
| Lower altitude | SYDD | *ABSP* | 27.516 | 88.99 | 3680 | 28 | 27.4±10.1 | 34 | 873 | 1938-2014 | 0.25 | 0.2 |

| Treeline | DJU | *ABSP* | 27.837 | 87.47 | 3918 | 19 | 25.0±5.8 | 132 | 344 | 1780-2006 | 0.40 | 0.17 |
|---|---|---|---|---|---|---|---|---|---|---|---|---|
| Middle altitude | DJM | *ABSP* | 27.838 | 87.46 | 3657 | 19 | 40.1±9.5 | 129 | 450 | 1774-2006 | 0.43 | 0.16 |
| Lower altitude | DJD | *ABSP* | 27.84 | 87.46 | 3378 | 13 | 28.3±6.6 | 85 | 556 | 1893-2006 | 0.37 | 0.18 |

*ABSP*, *Abies spectabilis*; rbar, the mean inter-series correlation; MS, mean sensitivity; DBH, diameter at breast height.

**Table S2.** Information for the tree-ring samples for isotope measurement on the southern Tibetan Plateau.

| Position | Site | Species | Trees | TRW time span | Isotope data time span | Mean age | DBH (cm) | rbar | MS |
|---|---|---|---|---|---|---|---|---|---|
| Treeline | SYDU | *ABSP* | 6 | 1784-2014 | 1901-2014 | 160 | 25.7±2.9 | 0.25 | 0.23 |
| Middle altitude | SYDM | *ABSP* | 5 | 1837-2014 | 1949-2014 | 90 | 28.7±4.9 | 0.25 | 0.24 |
| Lower altitude | SYDD | *ABSP* | 5 | 1938-2014 | 1973-2014 | 51 | 25.4±2.9 | 0.35 | 0.22 |
| Treeline | DJU | *ABSP* | 7 | 1780-2006 | 1888-2006 | 155 | 31.5±4.2 | 0.47 | 0.15 |
| Middle altitude | DJM | *ABSP* | 5 | 1774-2006 | 1869-2006 | 207 | 55.5±8.4 | 0.31 | 0.18 |
| Lower altitude | DJD | *ABSP* | 5 | 1893-2006 | 1897-2006 | 107 | 39.9±2.0 | 0.54 | 0.15 |

*ABSP*, *Abies spectabilis*; rbar, the mean inter-series correlation; MS, mean sensitivity; DBH, diameter at breast height.

*2. While the authors use ARSTAN with negative exponential or linear detrending, they do not explain how this affects long-term trends—the central focus of the paper. Detrending can remove real long-term growth signals (e.g., due to $CO_2$ fertilization). Were raw BAI series analyzed? Were alternative detrending methods tested (e.g., Regional Curve Standardization)? This needs clarification and discussion.*

**[Response]:** We appreciate the reviewer's valuable comment regarding the potential influence of detrending methods on growth trend detection. In this study, we used raw basal area increment (BAI) series to assess long-term growth trends, as BAI is less susceptible to biological trends and provides a more direct measure of stem biomass compared to tree-ring width data (Franco and Fares, 2008; Yang et al., 2022b;

Martinez-Sancho et al., 2018). To further minimize age-related effects, we excluded the early growth period from both trend analysis and isotope measurements.

For climate-growth relationship analysis, we applied standard detrending procedures to BAI series using both negative exponential/linear and spline function approaches (Conover, 2012; Van Der Sleen et al., 2015). Importantly, our results revealed consistent climate-growth relationships across different detrending methods for all major climate variables.

[Figure]

**Figure 5. Pearson correlation coefficients between basal area increment (BAI) and climatic variables, including (a) mean temperature, (b) total precipitation, (c) relative humidity and (d) standardized precipitation-evapotranspiration index (SPEI) on both monthly and seasonal scales for Himalayan fir during the period 1965–2013.** It should be noted that the BAI chronologies were derived exclusively from samples used for isotope measurements. "Grs" denotes the growing season (June to September). "Cor_value" and "abs_value" represent the correlation coefficient value and the absolute value of the correlation, respectively.

The results using a smoothing spline function with 50% variance cutoff at two thirds of the series length are as follows:

[Figure]

To enhance clarity, we have revised the Methods section to: (1) explicitly state that detrended data were used exclusively for climate-growth analysis; (2) clearly distinguish between the applications of raw BAI (for growth trends) and standardized BAI chronologies (for climate responses); (3) provide more detailed documentation of our detrending procedures. The newly added text is as follows: "The raw basal area

increment (BAI) series were used to assess long-term growth trends, as BAI is less susceptible to biological trends and provides a more direct measure of stem biomass compared to tree-ring width data (Franco and Fares, 2008; Yang et al., 2022b; Martinez-Sancho et al., 2018). To further minimize age-related effects, we excluded the early growth period from both trend analysis and isotope measurements. For climate-growth relationship analysis, we applied standard detrending procedures to BAI series using either a negative exponential or a linear function. The calculation and detrending of BAI was performed using the R package dplR (Bunn, 2008). ".

**3.** *The use of theoretical Ci/Ca trajectories (Scenarios 1 – 3) is oversimplified. These scenarios assume fixed relationships that rarely hold across environmental gradients or time. The authors treat alignment with Scenario 1 ("Ci = constant") as evidence of physiological strategy but offer no underlying reasoning in terms of stomatal control, leaf traits, or drought response. This needs much deeper physiological interpretation.*

**[Response]:** We sincerely appreciate the reviewer's insightful comment. We fully acknowledge that the theoretical scenarios of Ci/Ca changes presented in our initial analysis were overly simplistic and did not provide meaningful mechanistic explanations for our core findings. In response to this valuable feedback, we have removed both the Ci/Ca analysis and related discussion from the revised manuscript.

**4.** *The authors repeatedly attribute growth changes to rising Ca or warming based on correlations, without ruling out confounding variables (e.g., age, stand density, soil conditions). The SEM framework is promising but underdeveloped, and the model structure, diagnostics, and assumptions are not detailed.*

**[Response]:** Thank you for your valuable comment. We fully acknowledge that factors such as tree age, stand density, and soil conditions can influence tree growth. To minimize age-related effects, particularly in the early growth stages, we excluded tree rings from initial years showing strong growth trends, thus the shorter temporal coverage of isotope data compared to tree-ring width chronologies (Table S2). While

other stand characteristics (e.g., density and soil conditions) may indeed affect growth-climate-$CO_2$ relationships - and we have now included this information in the revised manuscript (Tables S1-S2) - these factors remain relatively stable over time and are unlikely to explain the observed temporal growth changes in recent decades. Rather, the pronounced climate change and rising atmospheric $CO_2$ concentrations appear to be the dominant drivers of interannual growth variations of trees in the high-altitude forests of this study (Panthi et al., 2020; Yang et al., 2022a).

To enhance the robustness of our structural equation modeling (SEM) approach, we have: (1) expanded the methodological description regarding factor selection in model construction; (2) clarified the interpretation of statistical outputs and model diagnostics; and (3) provided comprehensive piecewise SEM results in the supplemental materials (Table S3).

**Table S3.** Summary of the piecewise structural equation meta-model (pSEM) for testing the influences of climatic factors on basal area increment (BAI) and intrinsic water-use efficiency (iWUE) of Himalayan fir during the period 1965 – 2010s at different altitudes. $\beta$ is the standardized regression coefficient, and S.E. is the standard error. Abbreviations: Tpaut = previous autumn mean temperature; Tsum = summer mean temperature; Tspr = spring mean temperature; Tgrs = growing season mean temperature (June to September); RHspr = spring relative humidity; RHgrs = growing season relative humidity (June to September); RHwin = winter relative humidity; SPEIspr = spring standardized precipitation-evapotranspiration index; SPEIgrs = growing season standardized precipitation-evapotranspiration index (June to September).

| Site | Response variable | Predictor variable | $\beta$ | S.E. | Critical value | P-value |
|------|-------------------|--------------------|---------|------|----------------|---------|
| SYDU | BAI | iWUE | 0.1166 | 0.0309 | 0.7472 | 0.4589 |
|  | BAI | Tgrs | 0.1435 | 0.7033 | 0.8961 | 0.3751 |
|  | BAI | Tpaut | 0.3777 | 0.2276 | 2.8362 | <0.01 |
|  | BAI | Pgrs | -0.2081 | 0.0046 | -1.556 | 0.1269 |
|  | iWUE | Tgrs | 0.5308 | 2.7814 | 4.2362 | <0.001 |
|  | iWUE | Tpaut | -0.0738 | 1.1135 | -0.5724 | 0.5699 |
|  | iWUE | SPEIspr | 0.1567 | 1.8069 | 1.2155 | 0.2305 |
| SYDM | BAI | iWUE | 0.2137 | 0.0703 | 1.4584 | 0.152 |
|  | BAI | Tgrs | 0.2994 | 2.354 | 2.0129 | <0.1 |
|  | BAI | Tpaut | 0.1949 | 0.758 | 1.5834 | 0.1207 |
|  | BAI | Pgrs | 0.1959 | 0.0156 | 1.5546 | 0.1274 |
|  | BAI | RHgrs | -0.2499 | 0.5443 | -1.9459 | <0.1 |
|  | iWUE | Tgrs | 0.5562 | 4.2002 | 4.3683 | <0.001 |
|  | iWUE | Tpaut | 0.1356 | 1.5725 | 1.1065 | 0.2744 |
|  | iWUE | RHgrs | 0.0315 | 1.1155 | 0.2491 | 0.8044 |

| | | | | | | |
|---|---|---|---|---|---|---|
| | BAI | iWUE | 0.4527 | 0.0997 | 3.0971 | <0.01 |
| | BAI | Tgrs | 0.2431 | 2.8764 | 1.7251 | <0.1 |
| | BAI | Tspr | 0.161 | 1.1319 | 1.4095 | 0.1673 |
| SYDD | BAI | Pspr | 0.2083 | 0.0215 | 1.8744 | <0.1 |
| | iWUE | Tgrs | 0.5507 | 3.7033 | 4.4483 | <0.001 |
| | iWUE | Tspr | 0.2051 | 1.7453 | 1.7063 | <0.1 |
| | iWUE | RHgrs | -0.2113 | 0.9069 | -1.7321 | <0.1 |
| | BAI | iWUE | 0.0256 | 0.0323 | 0.174 | 0.8628 |
| | BAI | Tgrs | 0.2596 | 0.379 | 1.659 | 0.1056 |
| | BAI | Tspr | 0.1989 | 0.2468 | 1.2472 | 0.2202 |
| DJU | BAI | RHspr | 0.4223 | 0.0459 | 2.906 | <0.01 |
| | iWUE | Tgrs | -0.0401 | 2.0466 | -0.216 | 0.8302 |
| | iWUE | Tspr | 0.3189 | 1.2732 | 1.7613 | <0.1 |
| | iWUE | RHspr | 0.3172 | 0.2197 | 2.0745 | <0.05 |
| | iWUE | SPEIspr | 0.2289 | 1.9737 | 1.363 | 0.1811 |
| | BAI | iWUE | 0.4975 | 0.0532 | 2.8913 | <0.01 |
| | BAI | Tgrs | -0.0034 | 0.8131 | -0.0197 | 0.9844 |
| | BAI | Tspr | -0.4129 | 0.4904 | -2.5119 | <0.05 |
| DJM | BAI | RHspr | -0.5529 | 0.0866 | -3.8923 | <0.001 |
| | BAI | Pspr | 0.2686 | 0.0585 | 1.919 | <0.1 |
| | iWUE | Tgrs | 0.442 | 2.2004 | 3.0331 | <0.01 |
| | iWUE | Tspr | 0.271 | 1.4057 | 1.8595 | <0.1 |
| | BAI | iWUE | -0.1942 | 0.063 | -1.5072 | 0.14 |
| | BAI | Tgrs | -0.5024 | 0.7402 | -3.9003 | <0.001 |
| DJD | BAI | RHwin | -0.2553 | 0.0577 | -2.0357 | <0.05 |
| | iWUE | Tpaut | 0.2125 | 1.3755 | 1.1598 | 0.2532 |
| | iWUE | Tgrs | 0.1124 | 2.1529 | 0.6132 | 0.5433 |

**5.** *Many of the main findings—that iWUE increased, growth responded positively to warming in wet regions, and drought limits growth at lower elevations—are well established in the literature. The manuscript would benefit from a clearer articulation of what is truly new.*

**[Response]:** Thank you for your comment. Our study advances current understanding of tree-growth responses to environmental changes by examining a unique combination of two key gradients—altitude and moisture—in forests dominated by a single species (*Abies spectabilis*).

Specifically, our study provides novel insights in two key ways:

(1) Spatial Heterogeneity of $CO_2$ Fertilization Effects: By comparing wet and dry

regions, we demonstrate that both temperature and water availability acts as critical thresholds, determining whether elevated $CO_2$ enhances growth. This finding underscores that growth responses in high-altitude forests depend on the interplay between elevational gradients and regional moisture conditions.

(2) Dual Effects of Temperature on Growth: Using piecewise structural equation modeling (pSEM), we quantify contrasting roles of the increased temperatures in growth dynamics. At lower altitudes with water deficits, conservative stomatal regulation likely explains why improved intrinsic water-use efficiency (iWUE) did not consistently translate into growth benefits.

*6. The manuscript references nutrient availability as a potentially limiting factor (e.g., lines 54–56, 238), citing literature that suggests it can constrain the $CO_2$ fertilization effect. However, no nutrient data (e.g., soil N or P) are presented or analyzed. This creates a mismatch between the framing and execution of the study. Either include relevant data or remove unsupported speculation.*

**[Response]:** Thank you for your insightful comment. We agree that our original discussion regarding the effects of nutrient availability may have been overly speculative, particularly given the lack of direct evidence in our current study. Accordingly, we have removed this section and the associated references to better focus on our actual findings in the revised manuscript.

*7. While I believe that using AI to assist with scientific writing is acceptable, the manuscript shows clear signs of overreliance, including inconsistent use of abbreviations, redundancy of abbreviations, and occasional lapses in technical accuracy. The dataset presented is valuable, but the manuscript requires substantial revision to improve clarity, coherence, and scientific precision before it can be considered for publication.*

**[Response]:** Thanks for the comment. We have carefully examined the whole manuscript to cure the issues regarding the inconsistent use of abbreviations and the technical inaccuracies to ensure the precision and coherence of the text.

*Minor comments*

*1. Use species name in italic.*

**[Response]:** Sure. Done as suggested.

*2. More information on the ice core data are needed*

**[Response]:** Thanks for your comment. We have added more information on the source of the $CO_2$ data in the revised manuscript: "$CO_2$ concentration data was derived from a combination of the reconstructed values (period 1900-2003) using ice cores (Mccarroll and Loader, 2004; Boucher et al., 2014) and the direct observations of $CO_2$ concentration for the period 2004-2013, which were obtained from the Mauna Loa Observatory of America (http://www.esrl.noaa.gov/gmd/obop/mlo/)."

*3. Figures: Captions are too brief (especially in Figs. 2, 3, 4, 5) (also, there are no legend for coloring in figures).*

**[Response]:** Thanks for your comment. We have added the color legends in the figures and supplied more information in the captions for the figures 2-4. Note that the original Fig. 5 was removed in the revised manuscript.

[Figure]

**Figure 2. Temporal trends of basal area increment (BAI) for Himalayan fir at different altitudes during the period 1900–2010s.** The solid lines depict piecewise linear regression models, with a breakpoint at 1965. The model statistics include the slope of BAI ($\beta$, cm²year⁻¹), coefficient of determination ($R^2$) and associated p-values. The shaded regions denote 95% confidence intervals for the regression fits.

[Figure]

**Figure 3. Temporal trends of intrinsic water-use efficiency (iWUE) during 1900-2010s for Himalayan fir at different altitudes.** The solid lines depict piecewise linear regression models, with a breakpoint at 1965. The model statistics include the slope of BAI (β, cm² year-1), coefficient of determination ($R^2$) and associated p-values. The shaded regions denote 95% confidence intervals for the regression fits.

[Figure]

**Figure 4. Long-term relationships between intrinsic water-use efficiency (iWUE) and basal area increment (BAI) of Himalayan fir at different altitudes across two hydrologically distinct sites.** Linear regression results for each period are shown, along with the explained variance ($R^2$) and significance levels (p-values). Shaded areas represent 95% confidence intervals.

*4. Figure 4, it is hard to believe the results about iWUE since a lot of p-values shoz no significance of trends.*

**[Response]:** Thank you for your comment. Indeed, non-significant correlations were observed prior to 1965 for most sites, with the exception of the treeline site in the wet region (SYD) and the lower-altitude site in the dry region (DJ). However, we note gradual shifts in the correlations between iWUE and BAI across the two altitudinal gradients under differing hydrological conditions. Specifically, post-1965, the iWUE-BAI correlations transitioned from negative to positive with increasing altitude in the drier site (DJ), while the BAI-iWUE slope exhibited a progressive decline as

altitude rose.

[Figure]

**Figure 4. Long-term relationships between intrinsic water-use efficiency (iWUE) and basal area increment (BAI) of Himalayan fir at different altitudes across two hydrologically distinct sites.** Linear regression results for each period are shown, along with the explained variance ($R^2$) and significance levels (p-values). Shaded areas represent 95% confidence intervals.

*5. Any more information on the SPEI data? How were the data collected? Where they gridded data? Were the data extracted at specific locations?*

**[Response]:** Thanks for your comment. We have extracted the 3-month standardized precipitation evapotranspiration index (SPEI) data from the climate explorer (http://climexp.knmi.nl). The SPEI were calculated based on the climate data from the Climate Research Unit (CRU, University of East Anglia) TS Version 4.01 with a spatial resolution of 0.5°x 0.5°over the period from 1900 to 2014. The region mean values of SPEI calculated over the grids within the region 27.75°-28.25°N, 88.75°-89.25°E for SYD, and the region 27.95°-28.45°N, 86.75°-87.25°E for DJ, respectively.

**Reference**

Boucher, E., Guiot, J., Daux, V., Danis, P. A., and Dussouillez, P.: An inverse modeling approach for tree-ring-based climate reconstructions under changing atmospheric $CO_2$ concentrations, Biogeosciences,11,12(2014-06-17), 10, 3245-3258, 2014.

Conover, W.: The rank transformation—an easy and intuitive way to connect many nonparametric methods to their parametric counterparts for seamless teaching introductory statistics courses, Wiley Interdisciplinary Reviews: Computational Statistics, 4, 10.1002/wics.1216, 2012.

Farquhar, G. and Richards, R. A.: Isotopic Composition of Plant Carbon Correlates With Water-Use Efficiency of Wheat Genotypes, Functional Plant Biology, 11, 539-552, 10.1071/PP9840539, 1984.

Franco, B. and Fares, Q.: A Theory-Driven Approach to Tree-Ring Standardization: Defining the Biological Trend from Expected Basal Area Increment, Tree-Ring Research, 64, 81-96, 10.3959/2008-6.1, 2008.

Helama, S.: Expressing Tree-Ring Chronology as Age-Standardized Growth Measurements, Forest Science, 61, 817-828, 10.5849/forsci.14-139, 2015.

Linares, J. C. and Camarero, J. J.: From pattern to process: linking intrinsic water-use efficiency to drought-induced forest decline, Global Change Biology, 18, 1000-1015, 10.1111/j.1365-2486.2011.02566.x, 2012.

Liu, X., Zhao, L., Voelker, S., Xu, G., Zeng, X., Zhang, X., Zhang, L., Sun, W., Zhang, Q., Wu, G., and Li, X.: Warming and CO2 enrichment modified the ecophysiological responses of Dahurian larch and Mongolia pine during the past century in the permafrost of northeastern China, Tree Physiol, 39, 88-103, 10.1093/treephys/tpy060, 2019.

Martinez-Sancho, E., Dorado-Linan, I., Gutierrez Merino, E., Matiu, M., Helle, G., Heinrich, I., and Menzel, A.: Increased water-use efficiency translates into contrasting growth patterns of Scots pine and sessile oak at their southern distribution limits, Glob Chang Biol, 24, 1012-1028, 10.1111/gcb.13937, 2018.

McCarroll, D. and Loader, N. J.: Stable isotopes in tree rings, Quaternary Science Reviews, 23, 771-801, 10.1016/j.quascirev.2003.06.017, 2004.

McDowell, N. G.: Mechanisms linking drought, hydraulics, carbon metabolism, and vegetation mortality, Plant Physiol, 155, 1051-1059, 10.1104/pp.110.170704, 2011.

Panthi, S., Fan, Z.-X., van der Sleen, P., and Zuidema, P. A.: Long-term physiological and growth responses of Himalayan fir to environmental change are mediated by mean climate, Global Change Biology, 26, 1778-1794, 10.1111/gcb.14910, 2020.

Peters, R. L., Groenendijk, P., Vlam, M., and Zuidema, P. A.: Detecting long-term growth trends using tree rings: a critical evaluation of methods, Global Change Biology, 21, 2040-2054, 10.1111/gcb.12826, 2015.

Poyatos, R., Aguadé, D., Galiano, L., Mencuccini, M., and Martínez-Vilalta, J.: Drought-induced defoliation and long periods of near-zero gas exchange play a key role in accentuating metabolic decline of Scots pine, New Phytologist, 200, 388-401, 10.1111/nph.12278, 2013.

Pu, X. and Lyu, L.: Disentangling the impact of photosynthesis and stomatal conductance on rising water-use efficiency at different altitudes on the Tibetan plateau, Agricultural and Forest Meteorology, 341, 109659, https://doi.org/10.1016/j.agrformet.2023.109659, 2023.

Saurer, M., Siegwolf, R. T. W., and Schweingruber, F. H.: Carbon isotope discrimination indicates improving water-use efficiency of trees in northern Eurasia over the last 100 years, Global Change Biology, 10, 2109-2120, 2004.

van der Sleen, P., Groenendijk, P., Vlam, M., Anten, N. P. R., Boom, A., Bongers, F., Pons, T. L., Terburg, G., and

Zuidema, P. A.: No growth stimulation of tropical trees by 150 years of $CO_2$ fertilization but water-use efficiency increased, Nature Geoscience, 8, 24-28, 10.1038/Ngeo2313, 2015.

Voelker, S. L., Brooks, J. R., Meinzer, F. C., Anderson, R., Bader, M. K., Battipaglia, G., Becklin, K. M., Beerling, D., Bert, D., Betancourt, J. L., Dawson, T. E., Domec, J. C., Guyette, R. P., Korner, C., Leavitt, S. W., Linder, S., Marshall, J. D., Mildner, M., Ogee, J., Panyushkina, I., Plumpton, H. J., Pregitzer, K. S., Saurer, M., Smith, A. R., Siegwolf, R. T., Stambaugh, M. C., Talhelm, A. F., Tardif, J. C., Van de Water, P. K., Ward, J. K., and Wingate, L.: A dynamic leaf gas-exchange strategy is conserved in woody plants under changing ambient CO2 : evidence from carbon isotope discrimination in paleo and $CO_2$ enrichment studies, Glob Chang Biol, 22, 889-902, 10.1111/gcb.13102, 2016.

Wang, Z., Liu, X., Penuelas, J., Camarero, J., Zeng, X., Liu, X., Zhao, L., Xu, G., and Wang, L.: Recent shift from dominant nitrogen to $CO_2$ fertilization control on the growth of mature Qinghai spruce in China's Qilian Mountains, Agricultural and Forest Meteorology, 343, 109779, 10.1016/j.agrformet.2023.109779, 2023.

Wu, G., Liu, X., Chen, T., Xu, G., Wang, W., Zeng, X., and Zhang, X.: Elevation-dependent variations of tree growth and intrinsic water-use efficiency in Schrenk spruce (Picea schrenkiana) in the western Tianshan Mountains, China, Front Plant Sci, 6, 309, 10.3389/fpls.2015.00309, 2015.

Yang, R.-Q., Fu, P.-L., Fan, Z.-X., Panthi, S., Gao, J., Niu, Y., Li, Z.-S., and Bräuning, A.: Growth-climate sensitivity of two pine species shows species-specific changes along temperature and moisture gradients in southwest China, Agricultural and Forest Meteorology, 318, 108907, https://doi.org/10.1016/j.agrformet.2022.108907, 2022a.

Yang, R. Q., Zhao, F., Fan, Z. X., Panthi, S., Fu, P. L., Bräuning, A., Grießinger, J., and Li, Z. S.: Long-term growth trends of Abies delavayi and its physiological responses to a warming climate in the Cangshan Mountains, southwestern China, Forest Ecology and Management, 505, 119943, https://doi.org/10.1016/j.foreco.2021.119943, 2022b.

Zhang, X., Liu, X., Zhang, Q., Zeng, X., Xu, G., Wu, G., and Wang, W.: Species-specific tree growth and intrinsic water-use efficiency of Dahurian larch (*Larbc gmelinii*) and Mongolian pine (*Pinus sylvestris* var. *mongolica*) growing in a boreal permafrost region of the Greater Hinggan Mountains, Northeastern China, Agricultural and Forest Meteorology, 248, 145-155, 10.1016/j.agrformet.2017.09.013, 2018.